# Molecular Dynamics Study on Hugoniot State and Mie–Grüneisen Equation of State of 316 Stainless Steel for Hydrogen Storage Tank

**DOI:** 10.3390/ma16020628

**Published:** 2023-01-09

**Authors:** Li Yang, Huizhao Wang, Minghua Chi, Xiangguo Zeng, Yuntian Wang, Ping Zhao

**Affiliations:** 1MOE Key Laboratory of Deep Earth Science and Engineering, College of Architecture and Environment, Sichuan University, Chengdu 610065, China; 2China Construction Third Bureau Group Co., Ltd., 456 Wuluo Road, Wuhan 430064, China; 3City Construction College, Chengdu Polytechnic, Chengdu 610041, China

**Keywords:** intense shock loads, SC and PC 316 stainless steel, molecular dynamics, MSST, Mie–Grüneisen equation of state

## Abstract

To promote the popularization and development of hydrogen energy, a micro-simulation approach was developed to determine the Mie–Grüneisen EOS of 316 stainless steel for a hydrogen storage tank in the Hugoniot state. Based on the combination of the multi-scale shock technique (MSST) and molecular dynamics (MD) simulations, a series of shock waves at the velocity of 6–11 km/s were applied to the single-crystal (SC) and polycrystalline (PC) 316 stainless steel model, and the Hugoniot data were obtained. The accuracy of the EAM potential for Fe–Ni–Cr was verified. Furthermore, Hugoniot curve, cold curve, Grüneisen coefficient (*γ*), and the Mie–Grüneisen EOS were discussed. In the internal pressure energy-specific volume (*P*-*E*-*V*) three-dimensional surfaces, the Mie–Grüneisen EOSs show concave characteristics. The maximum error of the calculation results of SC and PC is about 10%. The results for the calculation deviations of each physical quantity of the SC and PC 316 stainless steel indicate that the grain effect of 316 stainless steel is weak under intense dynamic loads, and the impact of the grains in the cold state increases with the increase in the volume compression ratio.

## 1. Introduction

Two important problems facing the world this century are environmental protection and the energy crisis. Thus, humans must develop clean energy. Hydrogen energy has unique advantages in deep decarbonization and multi-energy complementarity to improve the stability of energy systems for energy conservation and emission reduction. It is regarded as an essential form of energy for achieving low-carbon societal development goals due to its high conversion efficiency, pollution-free fuel products, and renewability [1].

As a technologically mature alloy, 316 stainless steel is widely used in special industries, such as for high-precision equipment, medical equipment, and sophisticated weapons, because of its good corrosion resistance, non-magnetic property, work hardening, and high-temperature strength [2,3]. It is also an ideal material for making hydrogen storage tanks [4,5]. Due to the flammable/explosive characteristics of hydrogen, security has become one of the critical issues that restrict the popularization and application of hydrogen energy. Therefore, exploring the dynamic behavior of stainless steel in hydrogen storage tanks under extreme conditions of intense loads can provide a theoretical reference for hydrogen storage safety.

The load conditions of a hydrogen storage tank during use are very complex. In particular, the dynamic response, safety assessment, and structural design under intense shock loads attracted the attention of scholars. Ramirez et al. [6] used a specific continuum damage model to simulate the explosion process of gas storage tanks and obtained accurate predictions of the explosive pressure and explosive models of hydrogen tanks. Xu et al. [7] assessed the structural integrity of hydrogen storage tanks under impact loading and determined the material failure in each failure mode using finite element analysis. Shen et al. [8] considered and analyzed three damage modes of hydrogen storage tanks: shock waves, thermal radiation, and flying debris. They proposed a method for calculating the explosion energy in high-pressure hydrogen storage tanks, which provided the theoretical basis of taking safety measures to prevent explosions of hydrogen storage tanks. Lee et al. [9] studied the shock behavior of sintered 316L stainless steel from 10^−3^ s^−1^ to 7.5 × 10^3^ s^−1^, proposed a constitutive law that could predict the dynamic shock behavior of 316L stainless steel with high accuracy, and obtained the variation in the microstructure with the strain rate. This has a reference value for studying the strain rate effect of 316L stainless steel during impact.

The mechanical properties of metal materials under intense shock loads at high pressures are described by an equation of state (EOS). The Mie–Grüneisen EOS is one of the most widely used solid-state equations in explosion and shock dynamics, which establishes a relationship between the pressure (*P*), internal energy (*E*), and specific volume (*V*) for metal materials under extreme conditions, such as high pressures and temperatures [10]. The Hugoniot state and EOS are important for metal materials under extreme conditions of high pressures and temperatures [11,12]. In recent years, shock Hugoniot relationships that can be used as references in the development of EOSs have attracted significant attention [13]. To obtain the Hugoniot data of metal materials, a large number of computer simulations, experiments, and theoretical studies have been conducted, including first-principles theory and the classical mean field method. The research materials included Al, Cu, Pb, Ta, Mo, and W [14,15].

It is crucial to perform thorough research on the EOS and important characteristics such as the Hugoniot state. As a numerical simulation method that is an essential supplement to experimental research, methods based on molecular dynamics have attracted considerable attention in the study of the EOSs of metal materials. The multi-scale shock technique (MSST) [16] overcomes the shortcomings of equilibrium molecular dynamics (EMD) [17], non-equilibrium molecular dynamics (NEMD) [18], and first-principles methods [19], such as the high computational cost and small model size. MSST can effectively lower the calculation time and ensure accuracy in the simulation while ensuring a credible model size [20,21].

Many scholars have studied the properties of hydrogen storage tanks under strong impact loads and the Mie–Grüneisen EOS of many materials, and the relevant research has a reference value for studying material properties under strong impact loads. The superiority of the MSST method has been verified. However, research on the equation of state of 316 stainless steel in hydrogen storage tanks under intense shocks is lacking. There is a lack of micro-simulation approaches to study the dynamic behavior of hydrogen storage tanks. Furthermore, research on the grain effect of materials under intense shocks has been rarely reported. It is important to study the Hugoniot state and the Mie–Grüneisen EOS of 316 stainless steel and there are gaps and defects in the relevant research. Figure 1 shows the flowchart of this work. A micro-simulation approach based on the MSST method with less calculation time and higher accuracy was developed in this study, and the grain effect of material was considered and studied. The results of this study will help in revealing the dynamic behavior and the grain effect of 316 stainless steel for hydrogen storage tanks under extreme conditions of intense shock loads.

## 2. Methodology

### 2.1. Hugoniot Pressure (P_H_) and Internal Energy (E_H_)

The Hugoniot relation includes the mass conservation equation, the momentum conservation equation, and the energy conservation equation, which are shown as follows:(1)$$\rho \left(u_{s}-u_{p}\right)=\rho_{0}u_{s},$$
(2)$$P_{H}-P_{0}=\rho_{0}u_{s}u_{p},$$
(3)$$E_{H}-E_{0}=\frac{1}{2}(P_{H}+P_{0})(V_{0}-V),$$
where *ρ* is the material density, *u_s_* is the shock wave velocity, *u_p_* is the particle velocity, *P_H_* is the Hugoniot pressure, *E_H_* is the internal energy per unit mass, and *V* is the specific volume (*V* = 1/*ρ*). The subscript “0” indicates that the quantity is in an unshocked state, and “*H*” indicates that the quantity is in the Hugoniot state.

Within a wide high-pressure range, there is an approximately linear relationship between *u_s_* and *u_p_* for solid materials [22], expressed as follows:(4)$$u_{s}=C_{0}+\lambda u_{p},$$
where *C*_0_ is the volume speed of sound at zero pressure, and *λ* is a material parameter. Both *C*_0_ and *λ* can be determined by linear fitting of experimental or simulated *u_s_–u_p_* data.

When *P*_0_ = 0, by substituting the determined *C*_0_ and *λ* values into Equations (1)–(3), the Hugoniot pressure and its internal energy can be calculated as follows: (5)$$P_{H}\left(V\right)=\frac{\rho_{0}C_{0}^{2}\left(1-V/V_{0}\right)}{{\left[1-\lambda \left(1-V/V_{0}\right)\right]}^{2}},$$
(6)$$E_{H}=E_{0}+\frac{1}{2}P_{H}\left(V_{0}-V\right)=\int_{0}^{T}c_{v}dT+\frac{1}{2}P_{H}\left(V_{0}-V\right),$$
where *C_V_* is the specific heat at constant volume, *T*_0_ is the initial temperature, and *E*_0_ is usually ignored when the value of *E_H_* is much larger than that of *E*_0_.

### 2.2. Cold Pressure (Pc) and Cold Energy (Ec)

Similar to the case at 300 K, the shock wave velocity *u_s_*_0*K*_ at 0 K is also linearly related to the particle velocity *u_p_*_0*K*_, and the Hugoniot curve is expressed as [23]:(7)$$u_{s0K}=C_{0}^{\prime }+\lambda^{\prime }u_{p0K},$$
(8)$$P_{H0K}\left(V\right)=\frac{\rho_{0K}{C_{0}^{\prime }}^{2}\left(1-V/V_{0K}\right)}{{\left[1-\lambda^{\prime }\left(1-V/V_{0K}\right)\right]}^{2}},$$

$C_{0}^{\prime }$ and *λ*′ are the fitting parameters of the shock wave velocity *u_s_* and particle velocity *u_p_* at 0 K. They can be obtained through temperature correction based on *C*_0_ and *λ*. Based on the Hugoniot relation at 300 K (room temperature), *V*_0*K*_, *ρ*_0*K*_, $C_{0}^{\prime }$, *λ*′, and *F* are defined as follows [23]:(9)$$V_{0K}=V_{0}(1-\alpha_{v}T_{0}),$$
(10)$$\rho_{0K}=\rho_{0}(1+\alpha_{v}T_{0}),$$
(11)$${C_{0}^{\prime }}^{2}=C_{0}^{2}\frac{{\left(1-F\right)}^{2}}{{\left(1-F\lambda \right)}^{3}}[1-F\cdot \gamma (V_{0})+F\lambda ],$$
(12)$$\lambda^{\prime }=\lambda {\left(\frac{C_{0}}{C_{0}^{\prime }}\right)}^{2}\frac{{\left(1-F\right)}^{3}}{{\left(1-F\lambda \right)}^{4}}[(1+\frac{F\lambda }{2})(1-\frac{F\gamma (V_{0})}{2})-\frac{F\gamma^{2}(V_{0})}{4\lambda }(1-F\lambda )],$$
(13)$$F=\int_{0}^{300}\alpha_{v}(T)dT,$$
where *γ*(*V*_0_) is the Grüneisen coefficient without compression at 300 K (room temperature); *α_V_*(*T*) is the volume expansion coefficient, which is a material parameter that can be regarded as a constant when the temperature is in the range of 0–300 K, which can be obtained according to Equations (11) and (12), respectively.

The Born–Mayer (B–M) potential [24] and the Morse potential [25] are often used to describe the interaction between atoms at 0 K and high pressures. They can accurately describe the compressive properties of ionic crystals and metals. According to the relevant theory, the expressions of the cold pressure based on the B–M potential *P_c_*_-*BM*_ and the cold energy based on the B–M potential *E_c_*_-*BM*_ are defined as:(14)$$P_{c-BM}=Q\delta^{2/3}\{\exp[q(1-\delta^{-1/3})]-\delta^{2/3}\},$$
(15)$$E_{c-BM}=\frac{3Q}{\rho_{0K}}\{\frac{1}{q}\cdot \exp[q(1-\delta^{-1/3})]-\delta^{1/3}-(\frac{1}{q}-1)\}.$$

The cold pressure based on the Morse potential *P_c_*_-*M*_ and the cold energy based on the Morse potential *E_c_*_-*M*_ are defined as:(16)$$P_{c-M}=A\delta^{2/3}[e^{2B(1-\delta^{-1/3})}-e^{B(1-\delta^{-1/3})}],$$
(17)$$E_{c-M}=\frac{3A}{2\rho_{0K}B}{\left[e^{B(1-\delta^{-1/3})}-1\right]}^{2}.$$

In Equations (14)–(17), *A*, *B*, *Q*, and *q* are material constants; $\delta =V_{0K}/V$ represents the volume compression ratio; and *ρ*_0*K*_ is the density of the material at 0 K.

Generally, it can be assumed that the first and second derivatives of *P_H_*_0*K*_, isentrope *P_s_*, and *P_c_* are approximately equal at *P* = 0 and *T* = 0 [12,26], i.e.,
(18)$${\left(\frac{\partial P_{H0K}}{\partial V}\right)}_{V_{0K}}={\left(\frac{\partial P_{s}}{\partial V}\right)}_{V_{0K}}\approx {\left(\frac{\partial P_{c}}{\partial V}\right)}_{V_{0K}},$$
(19)$${\left(\frac{\partial^{2}P_{H0K}}{\partial V^{2}}\right)}_{V_{0K}}={\left(\frac{\partial^{2}P_{s}}{\partial V^{2}}\right)}_{V_{0K}}\approx {\left(\frac{\partial^{2}P_{c}}{\partial V^{2}}\right)}_{V_{0K}}.$$

With *C*_0_ and *λ* determined by simulations or experiments and Equations (5) and (12)–(17), the expressions of material constants *A*, *B*, *Q*, and *q* can be obtained as follows:(20)$$A=\frac{3\rho_{0K}{C_{0}^{\prime }}^{2}}{4\lambda^{\prime }-2},$$
(21)$$B=4\lambda^{\prime }-2,$$
(22)$$Q=\frac{3{C_{0}^{\prime }}^{2}\rho_{0K}}{q-2},$$
(23)$$q=6\lambda^{\prime }-3+\sqrt{3(12{\lambda^{\prime }}^{2}-20\lambda^{\prime }+9)}.$$

### 2.3. Grüneisen Coefficient (γ)

As a volume-related coefficient, *γ* reflects the relationship between the microscopic lattice vibrations and macroscopic thermodynamic properties of the material [27]. It is a significant physical quantity and can be expressed as [28]
(24)$$\gamma \left(V\right)=\frac{t-2}{3}-\frac{V}{2}\frac{d^{2}\left[P_{c}\left(V\right)V^{2t/3}\right]/dV^{2}}{d\left[P_{c}\left(V\right)V^{2t/3}\right]/dV},$$
where *t* represents the model for *γ*, and *t* = 0, 1, 2, corresponding to the Slater model *γ_s_*(*V*), Dugdale–MacDonald model *γ_DM_*(*V*), and free volume model *γ_f_*(*V*).

At the initial point (*P* = 0 and *V* = *V*_0_), Equation (24) can be approximated as
(25)$$\gamma \left(V_{0}\right)=-\frac{t+2}{3}-\frac{V_{0}}{2}\frac{P_{s}^{\prime \prime }\left(V_{0}\right)}{P_{s}^{\prime }\left(V_{0}\right)},$$

When *t* takes values of 0, 1, and 2, Equation (25) can be expressed as follows:(26)$$\gamma_{s}\left(V_{0}\right)=-\frac{2}{3}-\frac{V_{0}}{2}\frac{P_{s}^{\prime \prime }\left(V_{0}\right)}{P_{s}^{\prime }\left(V_{0}\right)},$$
(27)$$\gamma_{DM}\left(V_{0}\right)=-1-\frac{V_{0}}{2}\frac{P_{s}^{\prime \prime }\left(V_{0}\right)}{P_{s}^{\prime }\left(V_{0}\right)},$$
(28)$$\gamma_{f}\left(V_{0}\right)=-\frac{4}{3}-\frac{V_{0}}{2}\frac{P_{s}^{\prime \prime }\left(V_{0}\right)}{P_{s}^{\prime }\left(V_{0}\right)}.$$

At the initial point (*P* = 0 and *V* = *V*_0_), the first and second derivatives of *P_s_*(*V*) and *P_H_*(*V*) are the same. Based on the *P_H_* values obtained by simulations or experiments, $P_{H}^{\prime }$(*V*) and $P_{H}^{\prime \prime }$(*V*) can be obtained as follows:(29)$$P_{s}^{\prime }\left(V_{0}\right)=P_{H}^{\prime }\left(V_{0}\right)=-{\left(\frac{C_{0}}{V_{0}}\right)}^{2},$$
(30)$$P_{s}^{\prime \prime }\left(V_{0}\right)=P_{H}^{\prime \prime }\left(V_{0}\right)=4\left(\frac{C_{0}^{2}\lambda }{V_{0}^{3}}\right),$$
(31)$$\frac{P_{s}^{\prime \prime }\left(V_{0}\right)}{P_{s}^{\prime }\left(V_{0}\right)}=\frac{P_{H}^{\prime \prime }\left(V_{0}\right)}{P_{H}^{\prime }\left(V_{0}\right)}=-\frac{4\lambda }{V_{0}}.$$

By substituting Equations (29)–(31) into Equation (25), the following can be obtained: (32)$$\gamma_{s}\left(V_{0}\right)=2\lambda -\frac{2}{3},$$
(33)$$\gamma_{DM}\left(V_{0}\right)=2\lambda -1,$$
(34)$$\gamma_{f}\left(V_{0}\right)=2\lambda -\frac{4}{3},$$
(35)$$\gamma_{s}\left(V_{0}\right)=\gamma_{DM}\left(V_{0}\right)+\frac{1}{3}=\gamma_{f}\left(V_{0}\right)+\frac{2}{3}.$$

Some scholars have proposed a variety of empirical models of the volume-related Grüneisen coefficient *γ*, and most of them require the material parameters to be determined according to experimental data. This study was only based on numerical simulations without experimental conditions, so the available empirical formulas are [29,30]
(36)$$\gamma \left(V\right)=\gamma_{0}\frac{V}{V_{0}},$$
(37)$$\gamma \left(V\right)=\frac{2}{3}+\left(\gamma_{0}-\frac{2}{3}\right)\frac{V}{V_{0}}.$$

Based on the *λ* values determined from simulations or experiments, the Grüneisen coefficients of the three models under zero pressure can be obtained. The Grüneisen coefficients for different empirical equations can be obtained by Equations (36) and (37).

### 2.4. Mie–Grüneisen Equation of State

As a widely used EOS to describe the properties of solid materials under intense shock loads with extreme shock compression, the Mie–Grüneisen EOS established the relationship between the pressure, internal energy, and volume. In the classical form, the cold pressure and cold energy curves are used as reference curves, and the Hugoniot pressure curve and its internal energy curve obtained at room temperature can also be used as reference curves. The expressions of the two forms above are [31]
(38)$$P-P_{c}=\frac{\gamma (V)}{V}(E-E_{c}),$$
(39)$$P-P_{H}=\frac{\gamma (V)}{V}(E-E_{H}).$$

The Mie–Grüneisen EOS of 316 stainless steel can be obtained from Equations (38) and (39) based on the calculated results of the cold curves and the Hugoniot curves above.

## 3. Calculation Model

The embedded atom method (EAM) potential of Fe–Ni–Cr proposed by Zhou et al. [32] was used to describe the interactions between three main atoms of 316 stainless steel in the MSST simulation. This potential is suitable for austenitic stainless steel types, such as 316 stainless steel. The calculations were performed by the LAMMPS software [33].

The Fe–Ni–Cr austenitic alloy has a face-centered cubic (FCC) crystal structure. Some scholars have measured the lattice constant of 316 stainless steel with different methods, and the value is approximately 0.36 nm. Different measurement methods yield slightly different values, but the errors are within 1% [34,35]. Therefore, the lattice constant was set to 0.36 nm in this study. The molecular dynamics model was established to study the shock response of single-crystal (SC) and poly-crystalline (PC) 316 stainless steel. Both models were constructed with face-centered cubic crystal structures. To lower the calculation time and allow the MSST method to meet the accuracy requirements with a smaller model, the SC model size was set to 7.2 × 7.2 × 7.2 nm^3^, which contained 32,000 atoms (71% Fe–12% Ni–17% Cr). The PC model was established by the Atomsk software [36] with the Voronoi algorithm [37]. The size was set to 40.299 × 40.299 × 40.299 nm^3^, which contained 5,611,006 atoms (71% Fe–12% Ni–17% Cr) and 8 grains, and the average grain size was 25 nm. Both molecular dynamics models adopted periodic boundary conditions. Figure 2 shows the molecular dynamics model of the SC and PC 316 stainless steel.

The X-, Y-, and Z-axes correspond to the [100], [010], and [001] crystal orientations in the simulations, respectively. Before the shock wave was applied, the temperature was set to 300 K. Based on the NPT ensemble (constant number of particles, pressure, and temperature), a system equilibration was first performed to ensure the system was in a steady state for 10 ps. Subsequently, a series of shock waves were applied along the X-axis of the steady system, and the velocity was 6–11 km/s. The calculation time was set to be sufficiently long to ensure that the system was steady during the shock. That is, all the physical quantities, such as the temperature, pressure, and particle velocity, were in a steady state. The calculation time was set to 50 ps to obtain the particle velocity, pressure, temperature, and additional information.

## 4. Results and Discussion

### 4.1. Hugoniot Pressure (P_H_) and Internal Energy (E_H_)

In the simulation calculation, the value of *u_s_* was 6–11 km/s, which ensured the stability and effectiveness of the simulation. Table 1 shows the data of *u_p_* and *u_s_* for the 316 stainless steel obtained by the MSST method. There was an excellent linear relationship between the particle velocity and the wave velocity in the molecular dynamics simulation results. For the SC 316 stainless steel, the linear fit between *u_s_* and *u_p_* can be expressed as *u_s_ =* 4.906 + 1.443*u_p_*. The same shock wave velocity and other conditions of the single-crystal model were used to simulate the 25 nm polycrystalline model. The fit result was *u_s_ =* 4.826 + 1.454*u_p_*. The fitted lines are shown in Figure 3.

Hixson et al. [38] obtained the relation *u_s_* = 4.464 + 1.544*u_p_* through experiments. Table 2 shows the relative error between the simulation results of this study and the experimental results of Hixson et al. [38]. The results are very similar, which also verify the correctness of the simulations in this study and the accuracy of the EAM potential for Fe–Ni–Cr. The calculation results of the PC model are closer to the experimental results of Hixson et al. [38], because the PC model is more similar to the crystal structure of 316 stainless steel used in the experiment. It can be seen by comparing the calculated results of the SC and PC that the relative errors between the *λ* values are minimal (less than 1%), and the relative error between the *C_0_* values is approximately 1.63%. At the same shock wave velocity, the particle velocity of the PC is greater than that of the SC.

According to the linear relationship between the two speeds for the 316 stainless steel, the Hugoniot curves of 316 stainless steel can be determined using Equations (5) and (6). The red and blue curve represent results of SC and PC 316 stainless steel, respectively, while the green curve represents the results from Hixson et al. [38]. It can be seen that the internal energy increases slowly as the volume compression ratio (*V*/*V*_0_) increases at first, and then it increases rapidly after the volume compression ratio is greater than 1.2. The pressure decreases rapidly at first as the value of *V*/*V*_0_ increases, and then it decreases more slowly after the value of *V*/*V*_0_ is greater than 0.8. In Figure 4, the internal energy and pressure curves of the SC and PC 316 stainless steel almost overlap because the differences between *C_0_* and *λ* are very small. The deviations between the two increases as the volume compression ratio increases, but this change is very slight. The results of this paper are highly consistent with those of from Hixson et al. [38]. The maximum relevant error of *E_H_* is 3.4% at *V*_0_/*V* = 1.8, and the maximum relevant error of *P_H_* is 6.9% at *V*/*V*_0_ = 0.56.

### 4.2. Cold Pressure (Pc) and Cold Energy (Ec)

After *C*_0_ and λ were calculated, the values of the material constants of 316 stainless steel were obtained from Equations (9)–(13) and (18)–(23), as shown in Table 3. The cold energy *Ec* and cold pressure *Pc* of 316 stainless steel with the two microstructures are shown in Figure 5 and Figure 6, the results of data from Hixson et al. [38] are also presented in the figures. Both the Born–Mayer and Morse potentials were used to obtain the results. Both the cold energy and cold pressure increase with the volume compression ratio, and they increase faster at higher volume compression ratios (*V*_0K_/*V*). In particular, both the cold energy and cold pressure rise rapidly after *V*_0K_/*V* > 1.2.

The cold energy values calculated using the B–M and Morse potentials are very close, but the cold pressure values calculated by two different potentials deviate when *V*_0K_/*V* > 1.2, which increases with the volume compression. The cold energy and cold pressure calculated using the B–M potential are greater than those calculated using the Morse potential.

Comparing the calculation results of the SC and PC 316 stainless steel, it is found that the two are coincident at low volume compression ratios, and they deviate at large volume compression ratios. The cold energy values calculated by the B–M and Morse potentials separate after *V*_0K_/*V* > 1.2, and the cold pressure values deviate after *V*_0K_/*V* > 1.1. The deviations of the two physical quantities in the cold state between SC and PC 316 stainless steel increase conspicuously with the volume compression. The cold energy and cold pressure of SC 316 stainless steel are greater than those of PC 316 stainless steel.

Both the cold energy values based on the B–M potential and that based on the Morse potential are consistent with the results from Hixson et al. [38], which indicates that both the B–M and Morse potentials are suitable for calculating the cold energy. However, the cold pressure values based on the Morse potential deviate from the results from Hixson et al. [38] after *V*_0K_/*V* > 1.2, and those based on the B–M potential are consistent with the results from experimental data [38]. It indicates that both the B–M and Morse potentials are suitable for calculating the cold pressure when the volume compression ratio is low, but the B-M potential is more suitable for calculating the cold pressure.

### 4.3. Grüneisen Coefficient (γ)

Based on Equations (30)–(32) with the constant *λ* of 316 stainless steel obtained by linear fitting, the *γ*_0_ values of the Slater, Dugdale–Macdonald, and free-volume models were obtained. Subsequently, *γ*_0_(*V*) was substituted into Equations (36) and (37) to obtain the volume-related quantity *γ*(*V*). The *γ*(*V*) results calculated for the SC and PC 316 stainless steel were also used for comparison. The *γ*(*V*) results calculated by different empirical models are shown in Figure 7.

From Figure 7, it can be seen that with the increase in the volume compression ratio, the Grüneisen coefficient *γ*(*V*) shows a downward trend, and the rate of decrease is large at first and then gradually slows. When *V*_0_/*V* = 1.0, the values of *γ*_0_ are in the order of *γ_S_*(*V*_0_) > *γ_DM_*(*V*_0_) > *γ_f_*(*V*_0_). When using the same model, the results of Equations (36) and (37) are very close at the beginning of compression, and the deviation increases gradually during the process of compression. The *γ*(*V*) values based on Equation (36) are lower than that based on Equation (37). These results are similar to the results from simulation and experiment of Pb, Al, and Fe [12,39]. The calculated results for the SC and PC 316 stainless steel are very similar, and the deviations between the two decreases very slightly as the volume compression ratio increases. This is because the microstructure has less effect on lattice vibrations under high pressure. This result is the same as that of iron from Wang et al. [39]. The Grüneisen coefficient for the PC stainless steel is larger than that for the SC stainless steel, regardless of the models or equations used.

### 4.4. Mie–Grüneisen Equation of State

Based on Equations (38) and (39), the Mie–Grüneisen equation of state could be obtained. Equation (38) represents the cold curve as a reference, and Equation (39) represents the Hugoniot curve. The Hugoniot and cold curves were obtained by molecular dynamics simulations. According to the above analysis, the Grüneisen coefficient *γ_0_* was calculated by the Slater model, the volume-related quantity *γ*(*V*) was calculated by Equation (36), and the cold energy and cold pressure were calculated based on the B–M potential. The contours of the Mie–Grüneisen EOS are shown as internal pressure energy-specific volume (*P*-*E*-*V*) surfaces based on the Hugoniot curve and cold curve in Figure 8, where (a) and (b) correspond to the Hugoniot curve, and (c) and (d) correspond to the cold curve. The range of *E* in (a) and (b) is 0–15 MJ/kg, the range of *E* in (c) and (d) is 0–8 MJ/kg, and the range of *V*_0K_/*V* in (a)–(d) is 1–1.8. This corresponds to Figure 4, Figure 5 and Figure 6.

Figure 8a–d show the Mie–Grüneisen EOS results as *P*-*E*-*V* surfaces. Viewed from the front, the whole surface of the Mie–Grüneisen EOS is concave in the three-dimensional space. In Figure 8, it is intuitive that the pressure changes linearly with the energy, and the change with volume is more complicated. The pressure increases slowly at first and then rapidly with the volume compression. These two features correspond to Equations (38) and (39). There is a difference between the maximum pressures calculated based on the different reference curves. When the volume compression ratio is low, the pressure difference between the SC and PC 316 stainless steel is not large. This is because the calculation results of *P_C_*, *E_C_*, *P_H_*, and *E_H_* are not much different under low-pressure conditions. In the calculation results based on the two reference curves, the deviations between the pressures of the SC and PC 316 stainless steel increase as the volume compression ratio increases. Additionally, the selection of different reference curves affects the morphological characteristics of the *P*-*V*-*E* three-dimensional surface, as well as the value of the pressure with the same *E* and *V* values in the Mie–Grüneisen EOS, but the deviation is less than 1%. However, regardless of the reference curve used, the pressure of the SC 316 stainless steel is greater than the pressure of the PC 316 stainless steel with the same *E* and *V* values.

## 5. Conclusions

To understand the dynamic behavior of 316 stainless steel in hydrogen storage tanks under intense shock loads to promote the popularization and development of hydrogen energy, simulations of single-crystal and polycrystalline 316 stainless steel under intense loads were performed based on molecular dynamics simulations and multi-scale shock technology. The following conclusions were obtained:

(1) A micro-simulation approach with less calculation time and higher accuracy was developed in this study. The Hugoniot curve, cold pressure, cold energy, and Grüneisen coefficient of SC and PC 316 stainless steel were calculated. The *P*-*E*-*V* three-dimensional surfaces of the Mie–Grüneisen EOSs were obtained;

(2) The *u_s_–u_p_* relationships of both the SC and PC 316 stainless steel are highly consistent with the experiment results, which demonstrates the accuracy of the EAM potential for Fe–Ni–Cr;

(3) The use of the different reference curves influences the morphological characteristics and value of the pressure with the same *E* and *V* values in the Mie–Grüneisen EOS, but the error is less than 1%. The pressure of the SC 316 stainless steel is greater than the pressure of the PC 316 stainless steel with the same *E* and *V* values using both reference curves*;*

(4) The maximum error of the calculation results of SC and PC is about 10%, which indicates that the grain effect of 316 stainless steel is weak under strong dynamic loads. the impact caused by the grains in the cold state increases with the increase in the volume compression ratio:

(5) Grain effect cannot usually be ignored in the study of material properties, and the micro-simulation approach developed in this study and the method of research on grain effect can be used to explore the dynamic behavior of other materials.

## Figures and Tables

**Figure 1 materials-16-00628-f001:**
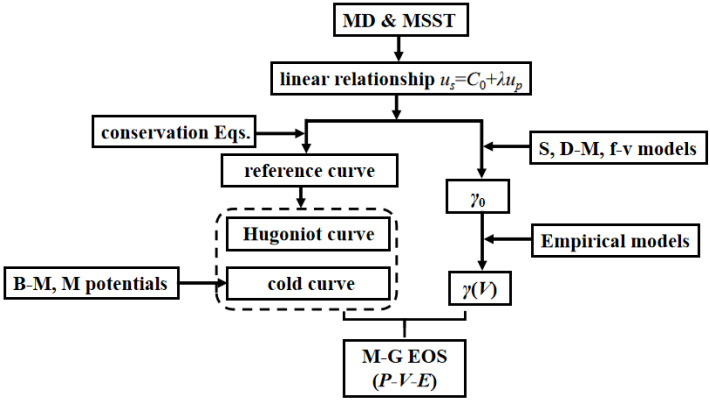
The flowchart of this work.

**Figure 2 materials-16-00628-f002:**
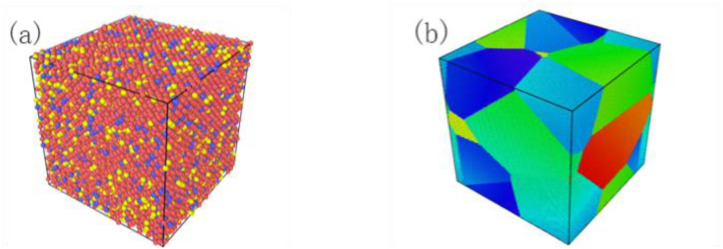
Molecular dynamics model of 316 stainless steel (**a**) single-crystal (SC) model (**b**)25 nm polycrystalline (PC) model.

**Figure 3 materials-16-00628-f003:**
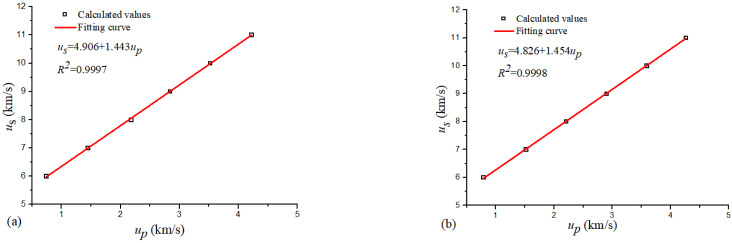
Fitting line of *u_s_* vs. *u_p_* for 316 stainless steel at 300 K: (**a**) SC model and (**b**) PC model.

**Figure 4 materials-16-00628-f004:**
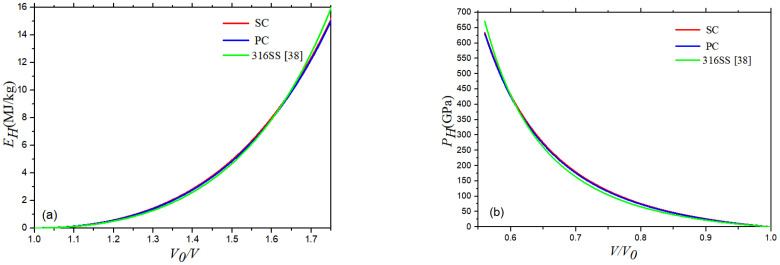
Hugoniot curves: (**a**) *E_H_* (*V*) curve and (**b**) *P_H_* (*V*) curve of 316 stainless steel at 300 K.

**Figure 5 materials-16-00628-f005:**
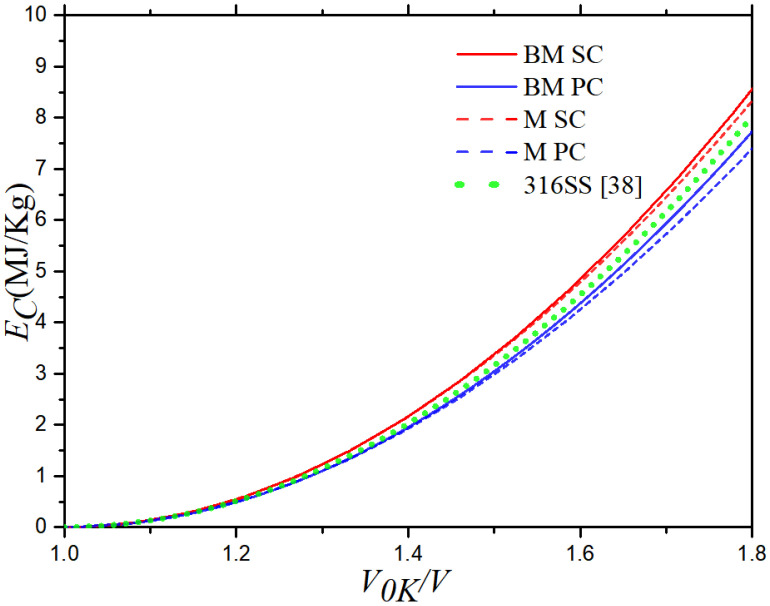
Cold energy curves of SC and PC 316 stainless steel.

**Figure 6 materials-16-00628-f006:**
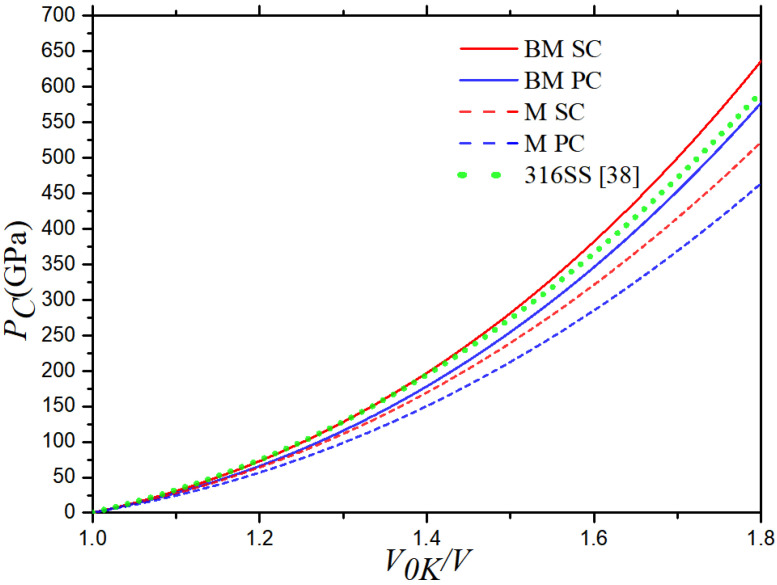
Cold pressure curves of SC and PC 316 stainless steel.

**Figure 7 materials-16-00628-f007:**
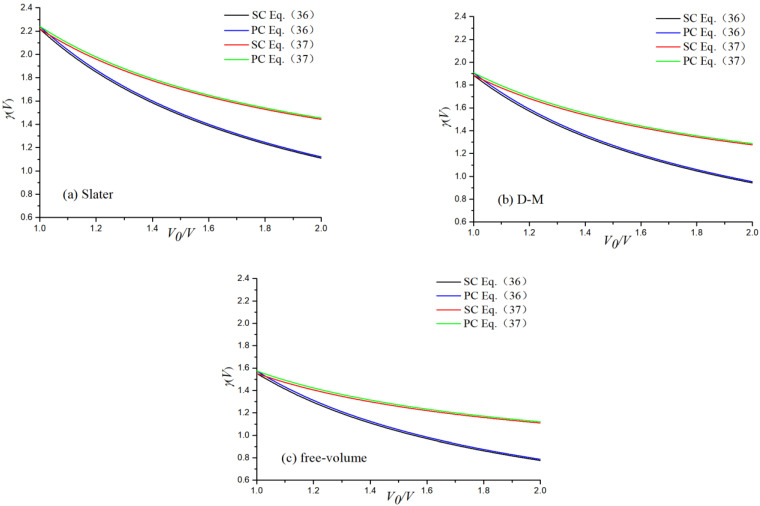
Grüneisen coefficient *γ*(*V*) based on three models.

**Figure 8 materials-16-00628-f008:**
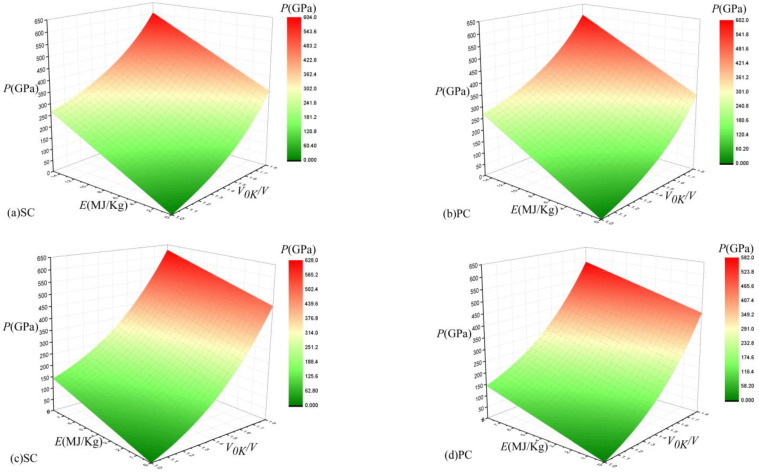
Mie–Grüneisen equation of state in *P*-*E*-*V* space, where (**a**,**b**) use the Hugoniot curve as a reference, and (**c**,**d**) use the cold curve as a reference.

**Table 1 materials-16-00628-t001:** Hugoniot data of 316 stainless steel at 300 K.

Material	Particle Velocity *u_p_* (km/s)	Shock Wave Velocity *u_s_* (km/s)	Material	Particle Velocity *u_p_* (km/s)	Shock Wave Velocity *u_s_* (km/s)
SC 316 stainless steel	0.74	6.00	PC 316 stainless steel	0.79	6.00
	1.45	7.00		1.52	7.00
	2.18	8.00		2.21	8.00
	2.84	9.00		2.90	9.00
	3.52	10.00		3.59	10.00
	4.22	11.00		4.26	11.00

**Table 2 materials-16-00628-t002:** Relative errors of Hugoniot parameters of 316 stainless steel.

Material	*C* _0_	*λ*	Material	*C* _0_	*Λ*
SC 316 stainless steel	9.01%	7.05%	PC 316 stainless steel	7.50%	5.83%

**Table 3 materials-16-00628-t003:** Material constants of 316 stainless steel.

	Specific Volume *V*_0k_ (cm^3^/g)	$C_{0}^{\prime }\;(\mathrm{km}/s)\;$	*λ*′	*A* (GPa)	*B*	*Q* (GPa)	*q*
SC	0.123	5.015	1.486	155.533	3.944	77.656	10.178
PC	0.123	4.933	1.498	148.678	3.992	72.213	10.219

## Data Availability

The data presented in this study are available on request from the corresponding author.

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
