# Peer review of "Molecular Dynamics Study on Hugoniot State and Mie–Grüneisen Equation of State of 316 Stainless Steel for Hydrogen Storage Tank"

_materials, 2023, doi:10.3390/ma16020628_

Round 1

Reviewer 1 Report

The authors presented an article « Molecular Dynamics Study on Hugoniot State and Mie–Grüneisen Equation of State of Stainless Steel for Hydrogen Storage Tank ». The authors are advised to consider the following comments for this paper.

·         316 stainless steel should be added to the title of the manuscript

·         Abstract

The abstract need to be improved. The abstract is written long and complicated. shortened and core findings of the study should be given. Please provide the main quantitative and qualitative research core findings. Demonstrate in the abstract novelty, practical significance. Briefly list the input and output parameters of the research.

·         Introduction

Seemingly, a comprehensive literature review was given. However, they were just summarized one- by-one. The authors have to stop after writing each example and think about the contributions and lack of knowledge for each paper. After that, in the final lines of the introduction give the blank spots of the topic. Then it will be clear what did authors make differently from the open literature.

In the last paragraph of the introduction section; What is the scientific novelty of the paper? What is the practical value? What makes this approach different from other researchers? Please specify. Gap and significance of the work must be included.

·         A flowchart for readers should be included.

·          Results and discussion

It is useful to add explanations of parameters to the results obtained. At least five sentences for each Figures. The results obtained should be explained by supporting the literature.

The equation of the curve fitted and R2 can be given on the Figure 2.

“There was an excellent linear relationship between the particle velocity and the wave velocity in the molecular dynamics simulation results.” Dynamic simulation conditions should be clearly stated.

Us values in Table 1 are rounded. Values should be written exactly.

The curves in Figures 2 a and b the same because the values are close. To avoid this, the values can be added on the figure.

Figure 6 shows that the results calculated for SC and PC are similar and the difference between them decreases as the volume compression ratio increases. The reasons for this should be discussed and compared with the literature.

Explanations and interpretations of Fig 7 figures c and d should be made in the text.

·         Conclusions

The conclusions need to be improved. The results are written long. It is necessary to more clearly show the novelty of the article. Add qualitative and quantitative results of your work. What is the difference from previous work in this area? Show practical relevance. What are the differences from previous works?

Suggestions should be made to increase the studies to be done in this field in conclusion section.

·         Authors should carefully study the comments and make improvements to the article step by step. All changes should be highlighted in color.

Author Response

Point 1:  316 stainless steel should be added to the title of the manuscript.

Response 1: Thank you for your comment. We have added it to the title of the manuscript.

Point 2: The abstract need to be improved. The abstract is written long and complicated. shortened and core findings of the study should be given. Please provide the main quantitative and qualitative research core findings. Demonstrate in the abstract novelty, practical significance. Briefly list the input and output parameters of the research.

Response 2: Thank you for your comment. Your comments have helped us greatly improve the abstract. We have shortened and simplified the abstract. The the main quantitative and qualitative research core findings were provided and the practical significance was added in the first sentence. We have demonstrated the novelty (the study of the grain effect under strong dynamic loads, the conclusion about the grain effect and the conclusion about the accuracy of the EAM potential for Fe-Ni-Cr) in the last two sentences. And we have listed the input parameters (the shock waves at the velocity of 6-11 km/s) and the output parameters (all the quantities and the Mie–Grüneisen EOS) of the research briefly.

Point 3:  Seemingly, a comprehensive literature review was given. However, they were just summarized one- by-one. The authors have to stop after writing each example and think about the contributions and lack of knowledge for each paper. After that, in the final lines of the introduction give the blank spots of the topic. Then it will be clear what did authors make differently from the open literature.

Response 3: Thank you for your comment. We have listed the advantages and disadvantages of the relevant research. And we have gave the blank spots of the topic (the lack of micro-simulation approach to study the dynamic behavior of hydrogen storage tanks and the rarity of the research on the grain effect of materials under intense shocks).

Point 4: In the last paragraph of the introduction section; What is the scientific novelty of the paper? What is the practical value? What makes this approach different from other researchers? Please specify. Gap and significance of the work must be included.

Response 4: Thank you for your comment. We have emphasized the novelty of this paper (superiority of this approach and the study of grain effect that fill the gap) and highlighted the differences between this research and other research as well as the advantages of this research. The significance of the work has also been included in the last sentence of the last paragraph.

Point 5: A flowchart for readers should be included.

Response 5: Thank you for your comment. We have added the flowchart in the end of the introduction, your comment helps us improve this paper.

Point 6: It is useful to add explanations of parameters to the results obtained. At least five sentences for each Figures. The results obtained should be explained by supporting the literature.

Response 6: Thank you for your comment. We have added explanations of the results. And there are at least five sentences for each Figures. We have compared with the results from the literature to support our results in section 4.

Point 7: The equation of the curve fitted and R2 can be given on the Figure 2.

Response 7: Thank you for your comment. We have added them on the Figure 2, the linear relationship can be clearly seen.

Point 8: There was an excellent linear relationship between the particle velocity and the wave velocity in the molecular dynamics simulation results.” Dynamic simulation conditions should be clearly stated.

Response 8: Thank you for your comment. But conditions of the molecular dynamic have been stated in the part 2(calculation model), and in our oppinion, it is not necessary to state the conditions here.

Point 9: Us values in Table 1 are rounded. Values should be written exactly.

Response 9: Thank you for your comment. But these values are exact values, they are not rounded. We set the values as 6, 7, 8, 9, 10, 11 km/s.

Point 10: The curves in Figures 2 a and b the same because the values are close. To avoid this, the values can be added on the figure.

Response 10: Thank you for your comment. I have added the equation of the two curves fitted on the Figure. The difference between them can be clearly seen.

Point 11: Figure 6 shows that the results calculated for SC and PC are similar and the difference between them decreases as the volume compression ratio increases. The reasons for this should be discussed and compared with the literature.

Response 11: We have interpreted the reason (the microstructure has less effect on lattice vibration under high pressure). And we compared it with the results from Wang, they are the same.

Point 12: Explanations and interpretations of Fig 7 figures c and d should be made in the text.

Response 12: Thank you for your comment. We have added the range of V0K/V in (c)– (d) in 4.4. The calculation method (figures c–d correspond to the cold curve) and the reference of the image range (corresponds to Figs. 3–5) of figures (c)– (d) are given in this paper. In the second paragraph expect for the last two sentences, there are the overall analysis and interpretation of the figures (a), (b), (c), and (d), including its morphological characteristics, the variation trends of pressure, and deviations of calculation results of SC and PC. All the explanations and interpretations of figures (a)-(b) and figures (c)-(d) are carried out together. And the same and different points of figures (a)-(b) and figures (c)-(d )are also discussed in the last two sentences.

Point 13: The conclusions need to be improved. The results are written long. It is necessary to more clearly show the novelty of the article. Add qualitative and quantitative results of your work. What is the difference from previous work in this area? Show practical relevance. What are the differences from previous works?

Response 13: We have shortened the results with listing core findings of the study. And we have clearly shown the novelty of the article and the differences from previous works (the superiority of this approach and the conclusion of grain effect that fill the gap). We have added the qualitative and quantitative results of my work (the relevant error and the properties of 316 stainless steel under intense shock loads). The practical relevance (promote the popularization and development of hydrogen energy) has been shown in the first sentence.

Point 14: Suggestions should be made to increase the studies to be done in this field in conclusion section.

Response 14: We have added the suggestions in the last point of the conclusion. In my oppinion, the micro-simulation approach developed in this study and the method of research on grain effect can be used in the studies in this field.

Reviewer 2 Report

1. In my opinion, there is a long way from such simulations to the popularization of hydrogen energy.

2. Would the simulation of such a material differ if it was used to store something else?

3. What are the practical conclusions for storage tanks? Is 316 stainless steel a good or weak for such storage tank?

Author Response

Point 1:  In my opinion, there is a long way from such simulations to the popularization of hydrogen energy.

 Response 1: As you commented, the popularization of hydrogen energy is difficult but important, it involves many fields and has many key problems to be solved. There is a long way from our simulations to the popularization of hydrogen energy.

 Point 2: Would the simulation of such a material differ if it was used to store something else?

 Response 2: Due to the flammable/explosive characteristics of hydrogen, we simulated the impact load on 316 stainless steel with MSST method based on the high pressure. If it was used to store something else like nitrogen, which is not prone to burning and exploding, the material will not be subjected to strong dynamic loads, then it is inappropriate to use this simulation method.

 Point 3:  What are the practical conclusions for storage tanks? Is 316 stainless steel a good or weak for such storage tank?

Response 3:  In our opinion, many other factors, such as corrosion resistance and fatigue performance, need to be considered when judging whether the material is suitable for the hydrogen storage tank. In terms of these points, 316 stainless steel is good for such storage tank. Because of this, there have been many hydrogen storage tanks are made of 316 stainless steel. However, hydrogen is flammable and explosive. So, we conducted this research to explore the dynamic behavior of 316 stainless steel for hydrogen storage tanks under extreme conditions of intense shock loads.

Reviewer 3 Report

The manuscript under consideration is “Molecular Dynamics Study on Hugoniot State and Mie–Grüneisen Equation of State of Stainless Steel for Hydrogen Storage Tank”. It is aimed forward understanding the dynamic behavior of 316 stainless steel for hydrogen storage tanks under extreme shock loads. The manuscript is written professionally and after some corrections is suitable for publishing in “Materials”.

These are several issues to be addressed.

1. Kindly check if “microscopic numerical simulation” is appropriate term (Abstract).

2. Scientific novelty and practical importance of research should be clearly stated in Abstract.

3. In the Introduction kindly check phrases “Hydrogen energy” and “essential form of energy”. Maybe “Hydrogen energetics” is more suitable?

4. What is meant by “renewability” (Introduction, 6th line)?

5. The next sentence is about 316 steel and it goes without any logical transition. Kindly consider beginning this sentence with a new paragraph.

6. Kindly rephrase to avoid using “issues” twice in one sentence (Introduction, the bottom of the first page).

7. The last paragraph of Introduction looks like conclusion section. Kindly reconstruct it: underline the outcomes from literature review and the goals of research.

8. In the Tables kindly indicate names of parameters, not just designating letters.

9. According to the Table 3, density of 316 steel is 8.13 g/sm3 (0.123 cm3/g). It seems to be unbelievably high value. Usually density of 316 steel is referenced as 7.98 g/cm3 (0.125 cm3/g). Kindly check.

10. In the Table 3 kindly check if “NC” is correct. Maybe “PC”?

Author Response

Point 1: Kindly check if “microscopic numerical simulation” is appropriate term (Abstract).

Response 1: Thank you for your comment. We have modified it into “micro-simulation approach”, which is appropriate term.

Point 2: Scientific novelty and practical importance of research should be clearly stated in Abstract.

Response 2: Thank you for your comment. Your comments have helped us greatly improve the abstract. I have demonstrated the novelty (the study of the grain effect under strong dynamic loads, the conclusion about the grain effect and the conclusion about the accuracy of the EAM potential for Fe-Ni-Cr) in the last two sentences. The significance of the work has also been clearly stated in the first sentence of the last paragraph.

 Point 3: In the Introduction kindly check phrases “Hydrogen energy” and “essential form of energy”. Maybe “Hydrogen energetics” is more suitable?

Response 3: Thank you for your comment, and we have checked these two phrases seriously. But we aimed to describe the advantages and necessity of hydrogen as an energy source here, so in our opinion, “Hydrogen energy” may be more suitable.

Point 4:  What is meant by “renewability” (Introduction, 6th line)? 

Response 4: It refers to the property of hydrogen energy as renewable energy (It means hydrogen energy can be reproduced by humans).

Point 5: The next sentence is about 316 steel and it goes without any logical transition. Kindly consider beginning this sentence with a new paragraph.

Response 5: Thank you for your comment. We have added a new paragraph at the beginning of this sentence according to your opinion. It is more logical.

 Point 6: Kindly rephrase to avoid using “issues” twice in one sentence (Introduction, the bottom of the first page).

Response 6: Thank you for your comment. we have modified the “safety issues” in the text into security. It makes the article read more smoothly.

 Point 7: The last paragraph of Introduction looks like conclusion section. Kindly reconstruct it: underline the outcomes from literature review and the goals of research.

Response 7: Thank you for your comment. We have underlined the outcomes from literature review and the goals of research just as your suggestion.

 Point 8:  In the Tables kindly indicate names of parameters, not just designating letters.

 Response 8: Thanks for your comment. Indicating names of parameters in the tables can help readers understand the contents of the table. We have added the names of us, up and V0K in the tables. However, λ, λ', A, B, q, and Q are material constant with no exact name. As for C0 and C0, they are volume speed of sound at zero pressure (300K and 0K), their names are two long, in our opinion, it is not appropriate to add them to the tables. So, we just indicated the names of us, up and V0K in the tables.

 Point 9: According to the Table 3, density of 316 steel is 8.13 g/sm3 (0.123 cm3/g). It seems to be unbelievably high value. Usually density of 316 steel is referenced as 7.98 g/cm3 (0.125 cm3/g). Kindly check.

Response 9: Thank you for your comment. But in the Table 3, V0k is the specific volume in 0K, it is obtained with eq.(7). And eq.(7) is widely used in the relevant research. The specific volume of 316 steel in 300K is 0.125 cm3/g, and the volume expansion coefficient of 316 steel is 4.56e-5/K.  V0K=Vo(1-αVT0)=0.125*(1-4.56e-5*300)=0.123cm3/g.

 Point 10: In the Table 3 kindly check if “NC” is correct. Maybe “PC”?

Response 10: We are sorry we made a mistake here. We have corrected it.